# Dietary Zinc and Risk of Prostate Cancer in Spain: MCC-Spain Study

**DOI:** 10.3390/nu11010018

**Published:** 2018-12-20

**Authors:** Enrique Gutiérrez-González, Adela Castelló, Pablo Fernández-Navarro, Gemma Castaño-Vinyals, Javier Llorca, Dolores Salas-Trejo, Inmaculada Salcedo-Bellido, Nuria Aragonés, Guillermo Fernández-Tardón, Juan Alguacil, Esther Gracia-Lavedan, Esther García-Esquinas, Inés Gómez-Acebo, Pilar Amiano, Dora Romaguera, Manolis Kogevinas, Marina Pollán, Beatriz Pérez-Gómez

**Affiliations:** 1Public Health & Preventive Medicine Teaching Unit, National School of Public Health, Carlos III Institute of Health, 28029 Madrid, Spain; e.gutierrez@isciii.es; 2Consortium for Biomedical Research in Epidemiology & Public Health (CIBER en Epidemiología y Salud Pública—CIBERESP), 28029 Madrid, Spain; acastello@externos.isciii.es (A.C.); pfernandez@isciii.es (P.F.-N.); gemma.castano@isglobal.org (G.C.-V.); javier.llorca@unican.es (J.L.); salas_dol@gva.es (D.S.-T.); isalcedo@ugr.es (I.S.-B.); nuria.aragones@salud.madrid.org (N.A.); gftardon@uniovi.es (G.F.-T.); juan.alguacil@dbasp.uhu.es (J.A.); esther.gracia@isglobal.org (E.G.-L.); esthergge@gmail.com (E.G.-E.); ines.gomez@unican.es (I.G.-A.); epicss-san@euskadi.eus (P.A.); manolis.kogevinas@isglobal.org (M.K.); mpollan@isciii.es (M.P.); 3Department of Epidemiology of Chronic Diseases, National Centre for Epidemiology, Carlos III Institute of Health, 28029 Madrid, Spain; 4Faculty of Medicine, University of Alcalá, 28871 Alcalá de Henares, Spain; 5ISGlobal, 08036 Barcelona, Spain; dora.romaguera@isglobal.org; 6IMIM (Hospital del Mar Medical Research Institute), 08003 Barcelona, Spain; 7Universitat Pompeu Fabra (UPF), 08002 Barcelona, Spain; 8Universidad de Cantabria—IDIVAL, 39011 Santander, Spain; 9Área de Cáncer y Salud Pública, FISABIO-Salud Pública, 46020 Valencia, Spain; 10Universidad de Granada—ibs.Granada, 18012 Granada, Spain; 11Cancer Epidemiology Section, Public Health Division, Department of Health of Madrid, 28035 Madrid, Spain; 12Oncology Institute IUOPA (Instituto Universitario de Oncología del Principado de Asturias), Universidad de Oviedo, 33003 Asturias, Spain; 13Centro de Investigación en Recursos Naturales, Salud y Medio Ambiente (RENSMA), Universidad de Huelva, 21004 Huelva, Spain; 14Department of Preventive Medicine and Public Health, Universidad Autónoma de Madrid and Idipaz, 28029 Madrid, Spain; 15Public Health Division of Gipuzkoa, BioDonostia Research Health Institute, 20014 San Sebastian, Spain; 16Balearic Islands Health Research Institute (IdISBa), University Hospital Son Espases, 07120 Palma de Mallorca, Spain; 17CIBER Fisiopathology of Obesity and Nutrition (CIBER-OBN), Carlos III Institute of Health, 28029 Madrid, Spain

**Keywords:** dietary zinc, prostate cancer, diet, genetic susceptibility

## Abstract

Zinc is a key trace element in normal prostate cell metabolism, and is decreased in neoplastic cells. However, the association between dietary zinc and prostate cancer (PC) in epidemiologic studies is a conflicting one. Our aim was to explore this association in an MCC-Spain case-control study, considering tumor aggressiveness and extension, as well as genetic susceptibility to PC. 733 incident cases and 1228 population-based controls were included for this study. Dietary zinc was assessed using a food frequency questionnaire, and genetic susceptibility was assessed with a single nucleotide polymorphisms (SNP)-based polygenic risk score (PRS). The association between zinc intake and PC was evaluated with mixed logistic and multinomial regression models. They showed an increased risk of PC in those with higher intake of zinc (Odds Ratio (OR) tertile 3vs1: 1.39; 95% Confidence interval (CI):1.00–1.95). This association was mainly observed in low grade PC (Gleason = 6 RRR tertile 3vs1: 1.76; 95% CI:1.18–2.63) as well as in localized tumors (cT1-cT2a RRR tertile 3vs1: 1.40; 95% CI:1.00–1.95) and among those with higher PRS (OR tertile 3vs1: 1.50; 95% CI:0.89–2.53). In conclusion, a higher dietary zinc intake could increase the risk of low grade and localized tumors. Men with higher genetic susceptibility might also have a higher risk of PC associated with this nutrient intake.

## 1. Introduction

Prostate cancer (PC) is the most frequent malignant neoplasm among European men, apart from non-melanoma skin cancer [1]. Unfortunately, neither the known genetic susceptibility [2,3] nor the main risk factors -age, race or familial history of PC- are modifiable, while the potential implications of environmental exposure in its aetiology, including the role of diet, remains uncertain [4]. Among the nutrients that have raised specific interest in this field, zinc occupies a major position. Even though skeletal muscles and bones account for approximately 90% of the total amount of zinc in the human body, the highest concentrations of this element are found in the prostate gland and fluids [5]. Zinc has an essential role in prostatic normal function. It inhibits the action of aconitase enzyme, that oxidates citrate to isocitrate in the mitochondria [6]; therefore, its accumulation in the acinar secretory epithelial cells of the peripheral zone of the prostate [7] allows the production and secretion of high levels of citrate, a major component of prostatic fluid [8].

Additionally, zinc is involved in the process of proliferation and apoptosis cell regulation in the prostate gland [8,9]. Zinc concentrations are consistently lower in PC tissue than in normal cells [6], probably as a result of a reduced expression of zinc transporters [8]. Besides, zinc depletion in PC tissue has been related to Gleason score and tumor aggressiveness [10,11]. As a consequence of zinc depletion, PC cells turn more energetically efficient while at the same time the pro-apoptotic-effect of zinc is reduced [7,11]. Furthermore, it has been proved that the treatment of neoplastic cells with zinc can inhibit cell proliferation, migration and invasion [6]. Therefore, it has been hypothesized that increasing zinc concentrations could promote the apoptosis of malignant cells [6,7]. 

Diet and supplements are the main source of zinc exposure (approximately 90%–95%) in general population, as the amount of zinc provided by drinking water is considered to be low (<10%) while the exposure through air, soil or occupation seems negligible [12,13]. However, the reports of epidemiological studies about zinc intake -dietary and/or in supplement- and PC risk have shown conflicting results, indicating that there is a need for more research on this issue to understand this complex relationship [14]. One possible reason for these contradictory results is the lack of data about tumor grade, as only a few previous studies have evaluated if this association was modified by tumor aggressiveness [15,16,17,18,19]. It is also relevant to note that none of these reports have assessed whether zinc might have a different effect among those with higher genetic susceptibility to developing PC.

In this study we explored the association between estimated dietary intake of zinc and risk of PC and tested whether tumor grade or extension, or genetic susceptibility to PC, measured with a polygenic risk core, might play a role in this relationship.

## 2. Materials and Methods 

### 2.1. Study Population and Data Collection

The subjects in this research were participants from the population-based multicase-control study MCC-Spain (www.mcc-spain.org). Briefly, this large multicenter study was designed to investigate the influence of environmental factors in common tumors or cancers with particular epidemiological characteristics in Spain (breast, prostate, colorectal, gastric and chronic lymphocytic leukemia) [20]. From 2008 to 2013, subjects from 12 different provinces of Spain were invited to participate. The inclusion criteria were to be aged 20–85 years, to be able to answer a questionnaire and to have resided in the study area for at least 6 months prior to recruitment. Incident histologically-confirmed cancer cases were identified and recruited through active search in surgical and oncological departments from collaborating hospitals. Population-based controls were randomly selected from the lists of General Practitioners (GP) at participating Primary Health centers within hospitals’ catchments areas. Individuals were invited telephonically to participate in the study in behalf of their GP. Controls subjects were frequency-matched to cases by age in five-year intervals, sex and study area. The study protocol was approved by the Ethics Committee of all participating centers (initial approval code: CEIC-IMAS 2008/3123/I), and each individual that agreed to participate signed an informed consent form prior to their enrolment.

A structured computerized questionnaire was administered face-to-face by trained interviewers to gather information on basic sociodemographic and lifestyle characteristics, lifetime residential history, self-reported anthropometric measures, occupational and medical history, drug intake, smoking status and physical activity, among others. Diet and alcohol consumption habits were collected by means of a semi-quantitative food frequency questionnaire (FFQ) previously validated in Spain [21], that was either administered during the interview or completed at home and returned by mail. The FFQ collected information of 140 items, including regional products and beverages, and was used to estimate specific nutrients groups during the previous year. Nutritional composition of foods were compiled from Spanish CESNID food composition tables [22] and used to estimate daily dietary zinc, calcium, energy and ethanol intake. In addition, we asked the participants whether they had ever been regular users of vitamins or dietary supplements (intake ≥ 1 year) as well as the brand of the specific products they had taken to identify those including zinc after assessing brand composition. 

For this study, we included incident PC cases (International Classification of Diseases 10th Revision: C61, D07.5) with no previous history of the disease, that had been diagnosed during the recruitment period in 14 hospitals from 7 Spanish provinces (Asturias, Barcelona, Cantabria, Granada, Huelva, Madrid and Valencia). The participation rate was 67.4% among cases and 52.2% among controls [20,23]. Those controls from provinces that had not recruited PC cases or with personal history of PC were excluded. A total of 1112 confirmed PC cases and 1493 controls were finally included in this study, of which 951 cases and 1298 controls had answered the FFQ on diet and had dietary estimates of zinc intake. Controls with prior history of adenoma surgery (*n* = 34), cases that had returned the questionnaire more than 6 months after diagnosis (*n* = 204), and those without BMI data (26 controls and 8 cases) were also excluded. Gleason grade [24] and tumor extension according to the American Joint Committee on cancer (AJCC) [25] were recorded for all prostate cancer patients. Only cases with Gleason ≥6 (i.e., International Society of Urological Pathology grading (ISUP) 1-5 [24]) were considered for this study. As a result, 733 cases and 1228 controls were included in this analysis.

The genotyping of the selected participants with available DNA (841 controls and 514 cases) was performed by the Centro Nacional de Genotipado (CEGEN-ISCIII) using the Infinium Human Exome BeadChip (Illumina, San Diego, USA) that includes 200,000 coding markers plus 5000 additional custom single nucleotide polymorphisms (SNP) selected from previous genome-wide association studies (GWAS) studies or in genes of interest. In Asturias and Huelva provinces, participants did not have DNA due to logistical reasons, and were not included in these analyses. We constructed a polygenic risk score (PRS), as explained in detail elsewhere [2]. In summary, we used Genomebrowser [26] to identify those genetic variants associated with PC through GWAS in the population with European ancestry, with PC as “reported trait” and a p-value threshold of 5 × 10^8^. We could finally locate in our data 56 of these SNPs. The PRS score was obtained by adding the number of copies of the risk allele of each SNP, weighted by their beta coefficients, obtained from logistic regression analyses [2].

### 2.2. Statistical Analysis

First, a descriptive analysis to determine general characteristics of the study population was performed. Dietary zinc was categorized in tertiles on the basis of the distribution among controls. We evaluated the association of zinc with PC cancer risk by fitting two mixed logistic regression models, both including province of residence as a random effect: (a) a basic model, which considered only design-derived variables (age, education) and the most commonly accepted risk factors for PC (BMI one year before diagnosis, family history of prostate cancer) as fixed effects; and (b) an adjusted model, which added total grains and legumes consumption as well as dietary calcium intake. In this context, grains and legumes are used as surrogates of dietary phytate [27], which is known to interfere with zinc absorption [28,29]. Calcium may also inhibit zinc absorption [29], while diets rich in calcium have been classified as probable risk factors for PC according to the World Cancer Research Fund [30]. As sensitivity analysis, included as supplementary material (Appendix A), we tested other models, taking into account energy intake (kcals/day), physical activity (mets/day), smoking one year before diagnosis (never, ex-smokers and smokers), alcohol consumption (g/day) and supplement intake (never user or ever users supplements without zinc, users of unspecified supplements, ever user of multivitamin or supplement brands with zinc). In addition, we also explored non-linear associations between total zinc intake and PC risk using restricted cubic spline models with knots at 10th, 50th, and 90th percentiles in all participants, and including the confounders of the adjusted model.

Afterwards, we assessed the association with PC risk considering tumor aggressiveness with multinomial logistic regression models, also adjusted by the variables described above. Cases were divided into low (Gleason = 6, equivalent to ISUP = 1) and high grade (Gleason > 6 or ISUP 2-5); and based on clinical extension at diagnosis cases were categorized in cT1-cT2a (localized) or cT2b-cT4. We also explored this association with other high-grade definitions (ISUP ≤ 2 vs. 3–5; AJCC 8th ed I-IIA vs IIB-IV) (Appendix A) as well as the possible confounding role of having had any PSA test in the last 5 years (Appendix A). Finally, we evaluated the role of genetic susceptibility in the relationship between zinc and PC by testing the corresponding interaction term, and by performing subgroup analyses based on the distribution of PRS tertiles. All the statistical analyses were performed with Stata 14, establishing statistical significance at two-sided *p* < 0.05.

## 3. Results

The main features of participant cases and controls are shown in Table 1, which also summarizes the characteristics of the genotyped subsample. There were no relevant differences regarding age, BMI or daily energy intake between groups. Dietary zinc intake, estimated by FFQ, ranged between 2.5 and 29.5 mg/day and was slightly higher in cases, with a borderline statistical significance. Use of supplements was very unusual in both groups, and the proportion of ever users was similar between them. Compared to controls, PC cases had lower educational level, reported higher alcohol consumption and had a higher proportion of family history of PC.

### 3.1. Zinc Intake and Prostate Cancer Risk

Table 2 presents the ORs of the relationship between estimated dietary zinc and PC. Men in the highest tertile of zinc intake were at higher risk of PC, (Odds ratio (OR): 1.27; 95% confidence interval (CI): 1.00–1.61), with a marginally significant trend (*p* = 0.053). When dietary calcium and grains and legumes intake were also considered, the magnitude of the association increased (OR: 1.39; 95% CI: 1.00–1.95; *p*-trend = 0.051). Sensitivity analysis adjusting by energy intake, physical activity, smoking, alcohol consumption and supplements did not show differences with previous findings (Appendix A).

When the dose–response relationship between dietary zinc intake and PC risk was modelled using restricted cubic splines, we observed an initial increase of risk with estimated intake of zinc, which was followed by a subsequent stabilization starting around 10 mg/day (Figure 1).

### 3.2. Zinc Intake and Prostate Cancer Risk by Tumor Aggressiveness and Extension

When we evaluated the association by aggressiveness, we observed that the increase in PC risk with zinc was mainly observed in low grade tumors (Gleason = 6 Adjusted relative risk ratio (RRR) tertile 3vs1: 1.66; 95% CI: 1.07–2.57) as well as in localized tumors (cT1-cT2a RRR tertile 3vs1: 1.40; 95% CI: 0.97–1.99) but not in those with Gleason score >6 or in advanced tumors (cT2b-cT4). Notwithstanding, the test for heterogeneity was not statistically significant in either case (Table 3). Similar results were obtained with other clinical classifications of tumors (ISUP [24]; AJCC 8th ed [25]) (Appendix A) or when PSA screening was included as an additional confounding factor (Appendix A).

### 3.3. Zinc Intake, Prostate Cancer Genetic Susceptibility and PC Risk

We also tested if the effect of dietary zinc was modified by the genetic susceptibility to PC, estimated with a previously designed polygenic risk score [2]. There were significant differences in this association by tertile of PRS (*p*-heterogeneity = 0.000): in those with lower genetic predisposition to PC there was no increased risk of zinc exposure, contrasting with the relationship found in those in the highest tertile (Table 4), and this probably not statistically significant due to low statistical power.

## 4. Discussion

Our results suggest that dietary zinc is related to an increased risk of PC, but also indicate that this association is stronger in low grade and localized prostatic malignancies than in aggressive PC. In addition, they highlight the possible modulator role of genetic susceptibility in this relationship, as the effect of zinc is mostly observed in those men with higher genetic predisposition towards this tumor.

The role of zinc on PC is still not well understood. On the one hand, in vitro and in situ evidence is quite consistent. Zinc concentrations are much lower in PC cells than in normal prostatic tissue [6,31], probably due to the local downregulation of zinc transporters, mainly ZIP1 [8,32], and it has been hypothesized that increasing zinc concentrations could promote the apoptosis of malignant PC cells [6,7]. In this context, some authors support the study of zinc as a possible preventive option against PC [6]. On the other hand, it seems that the association between dietary intake and the tightly regulated plasmatic levels of zinc is not very direct [33,34] and the complex homeostasis of this micronutrient makes it difficult to estimate to what extent increases in dietary zinc translate into higher levels in prostate cells, and whether they actually have beneficial effects in regard to PC risk.

In fact, epidemiologic studies on this issue have yielded quite conflicting results. Instead of reporting a preventive effect, some case-control [35,36,37] or cohort studies [15] found that zinc was associated with PC, as we describe here. Thus, a case-control on diet and prostate cancer in Hawaii was the first to show an increase of risk with zinc intake (diet plus supplements) that was statistically significant in men >70 years [36]. Later, Gallus et al. in Italy found that men in the highest quintile of dietary zinc were at the highest odds of PC [35]. Regarding supplement use, Zhang et al. also reported an increased PC risk in men taking supplements containing zinc for more than 5 years, although models further adjusted by the use of other mineral and vitamin supplements only showed an increased risk for those taking supplemental zinc for more than 10 years [37]. In the Health Professionals Follow-up study, the excess of risk was limited to men with zinc intakes higher than 100 mg/day and to those taking zinc supplements longer than 10 years [15].

In contrast, other researchers have described a protective effect of zinc on PC risk. In this sense, intake of supplements containing zinc was associated to a reduction in the risk of PC in a trial from Canada, but this effect was restricted only to those with normal PSA at baseline [38]. Kristal et al. reported a borderline inverse association between PC risk in men using zinc supplements daily, but not in those with lower frequency intake [18]. Also a VITAL cohort study found that supplementary zinc intake higher than 15 mg/day was related to a decreased PC risk among those with high vegetable consumption, while the opposite effect was observed among those with a lower intake of vegetables [19]. Finally, other case control-studies have failed to show any relationship between zinc intake (diet +/− supplements) and PC [16,17,39,40,41,42].

A recent meta-analysis has summarized 17 studies testing this association, showing an almost significant increase of risk among those with higher intake or higher levels (nails/hair/serum) (OR _high vs. low_ = 1.07; 95% CI: 0.98–1.20), which is consistent with our findings [41]. Focusing only on studies evaluating dietary zinc, which are more comparable to ours (different bioavailability and higher doses in supplements, usually accompanied by other minerals and vitamins), the meta-analysis obtained a similar estimate (OR _high vs. low_ = 1.05; 95% CI: 0.93–1.20) [41]. In addition, its dose-response trend [41] was also similar to the one presented in Figure 1. It is also noteworthy to observe that the non-lineal response we found also somehow mimics the published asymptotic regression of absorbed zinc and ingested zinc [43].

With regard to the plausibility of the observed results, the possible biological pathways involved remain unclear, although several hypotheses have been proposed. Some authors argue that zinc may have a boosting effect in telomerase activity [44], which is increased in malignant prostate cells [45]. Others point to the impairment of immune response, although this effect has only been observed at very high zinc intakes [46], while in contrast, there is growing and solid evidence of the crucial role of zinc in the immune system [47]. Another possible explanation focuses on insulin-like growth factor I (IGF-I). Zinc intake is correlated to IGF-I [48], while its levels [49] as well IGF-I-related genetic variants, are associated to PC risk [50]. Also, experimental models in TRAMP mice have shown a higher weight of prostate tumors, accompanied by higher IGF-1 serum level with both zinc-deficient and zinc-enriched diet, in contrast to standard zinc diet [51], pointing to a U-shaped relationship between this micronutrient and PC.

It is relevant to note that our participants mostly report medium-low zinc exposure through diet (see Figure 1), suggesting that the association between zinc and prostate cancer can occur at levels quite lower than the recommended intake. The mean dietary intake in both controls and cases was slightly higher than that reported for Spanish men between 45–64 years in the 2011 Spanish National Dietary Intake Survey [52], and for men between 65–75 years in ANTIBES study for 2012 [53]. However, in almost 50% of our study population, zinc intake did not meet the Estimated Average Requirements (EAR) (9.4 mg/day for men) of the Institute of Medicine [43]. Furthermore, in around 70%, the estimated intake was below the Recommended Dietary Allowance (RDA) (11 mg/day for men), and only in 5% of them intake was close to the Reference Dietary Intakes suggested by the Spanish Federation of Nutrition, Food and Dietetics Societies (15 mg/day for men) [54]. In addition, in MCC-Spain the rate of having ever used supplements is very infrequent (around 3% for cases and controls), which is somewhat lower than the results reported in other studies in Spain (8% in adults 18–64 years in 2002 [55]).

A very interesting and novel finding of this study is the differential effect of zinc intake depending on the genetic susceptibility of men to PC. Our data indicate that the possible deleterious effect of higher levels of zinc in the diet might be limited to men with higher PRS scores. This aspect has not been taken into account in any other study and might provide a new element for understanding the heterogeneous results reported. In addition, if this result is confirmed by other authors, it would point to the need to deepen into a more personalized approach in dietary intake recommendations for this nutrient.

One of the key questions in these analyses is how accurately can dietary estimates reflect the real exposure to zinc. In this sense, there are several known factors that can modulate the bioavailability of dietary zinc, including the source that provides this micronutrient. In our study, the food item with the highest correlation with zinc intake estimation is red meat followed by other animal foods (Appendix A); in fact, the Spanish adult population’s intake of meat, fish and their derivates account for 40% of their dietary zinc [52]. Some studies report that zinc absorption is enhanced when the source is animal food [56] which might be behind the more marked associations we found in our study. Another important source of zinc is dairy products; however, they are also rich in calcium that, as we have stated, might hamper zinc absorption and can be a risk factor for PC [30]. Thus, calcium might bias the estimation of zinc if it is not considered as possible confounder. Finally, the most relevant factor is the amount of dietary phytate, due to its known interference with zinc absorption [28,29]. Cereals, and also legumes, nuts and grains are the main sources of this compound [27]. However, none of the published studies up to date have explicitly incorporated phytate or any proxy of its intake in their analyses. In our case, as we did not have data on phytate intake, we used cereal, rice and legume intake as a proxy of this exposure, in order to get closer to real absorbable zinc. The inclusion of this factor, together with dietary calcium, as possible confounders in our models resulted, as expected, in an increase in the risk of PC with zinc levels.

Another intriguing result is the higher association with low grade or localized tumors. A possible reason for different effects of zinc intake by tumor stage might be a lower susceptibility to changes in zinc exposure in abnormal PC cells. Zinc depletion in PC tissue occurs both in low and high grade tumors [14], but the alteration of zinc transporters occurs in an early stage of PC progression [32,57] and the downfall of zinc in the tissue is also related to Gleason score and tumor aggressiveness [10,11]. An alternative possibility would be a role of zinc in slowing progression of PC. This hypothesis would explain the decreased risk of mortality by PC with dietary zinc reported in a Swedish cohort study, especially in men initially diagnosed with localized disease (T1-T2/M0)[58]. In this sense, it is interesting to note that modest increases in dietary zinc have been found to reduce DNA damage and to alter the concentrations of serum proteins involved in DNA repair, oxidative stress, and inflammation, without changes in the plasmatic levels of the element [34].

Only a few studies [15,16,17,18,19,59] on zinc intake and risk of PC have taken into account tumor aggressiveness. In general, they do not seem to agree with our findings, although the comparison is difficult as, according to current consensus in urology [24,60], this study included only PC cases with Gleason grade ≥ 6; lower grades are no longer considered PC. For dietary zinc, the Italian case-control found a stronger association for advanced disease (Gleason > 7) [35]. Also a case-control in Utah (USA) reported a peculiar trend, with high risk of aggressive PC –undifferentiated grade or regional/distance stage-around 13–16 mg/d but not for higher consumption [16], and another Swedish population-based case-control observed an association in advanced PC (T3-T4 or M1), with intakes over 9.1 mg/d, although the excess of risk disappeared after adjusting by energy intake [17]. While none of those studies gave risk estimates for non-aggressive PC, in both cases the association for advanced tumors was stronger than in total PC. Regarding supplement users, the Health Professionals Follow-up cohort study, in USA, found an increase of risk with zinc intakes ≥ 24 mg/day just in advanced cases (T3b-T4, M1), and not in localized tumors (T1b-T2b) [15]. However, it is curious to find that, among those studies pointing to lower risk with zinc intake, with independence of its statistical significance, the effect was also usually more pronounced for advanced tumors. This was the case with dietary zinc in the Prostate Cancer Prevention Trial (Gleason 8–10), although in this study there was not any association with zinc from supplements [59]. In contrast, risk of PC with regional/distant invasion was inversely associated to zinc from supplements in a Washington population-based case-control study which evaluated 7 doses/week versus none [18], and in a VITAL cohort, USA, that tested the effect of 10-yr supplemental zinc use [19].

One of the strengths of our study is the population-based design and the relatively large sample size of MCC-Spain, as well as the availability of histopathological information and genetic data. This information has allowed us to provide a stratified analysis by tumor aggressiveness and extension, and to test the interaction with genetic susceptibility. Another strong point is the estimation of zinc intake based on national food composition tables [22], with include data of local food items, and the use of FFQ, which has been considered an acceptable instrument for dietary assessment for zinc [61,62]. Notwithstanding, retrospective FFQs may always have inaccuracies and, in case-control studies, they can have recall bias, which might be differential according to case/control status. Based on the limitations of FFQs, some authors have suggested the use of biomarkers, like toenails, for a better integrated assessment of zinc exposure [63], although at this moment there is no good and generally accepted biomarker.

Among the limitations of our work, we cannot rule out a possible selection bias on the basis of a higher participation in cases compared to controls, although we have tried to compensate this problem by fitting multivariable models including educational level, which might be the most relevant differential factor between both groups. Also, dietary zinc represents an estimation of zinc exposure through diet but does not take into account other possible sources of zinc intake. Moreover, we do not have information on zinc metabolism or bioavailability. We have tried to include a proxy of phytate intake in our estimation, and have also contemplated the possible confounding role of calcium, but, obviously, we cannot rule out residual confounding by factors not yet identified, associated to both zinc intake and PC. Another limiting point is our lack of information about which cases were diagnosed because of PSA screening, as men prone to go to the urologist’s office for check-up might be more health-conscious and have a different dietary pattern. However, in our study, reporting at least one PSA test in the last 5 years was not associated to differences in zinc intake, and the inclusion of this factor in the models did don’t change the results.

## 5. Conclusions

High dietary zinc intake could be associated with a higher risk of PC, especially for low grade and localized tumors. Also, men with higher genetic susceptibility may be those at higher risk of PC associated with this nutrient. Based on our results, dietary zinc recommendations should consider genetic backgrounds. Further research is needed to confirm our findings and to elucidate the possible mechanisms of this association.

## Figures and Tables

**Figure 1 nutrients-11-00018-f001:**
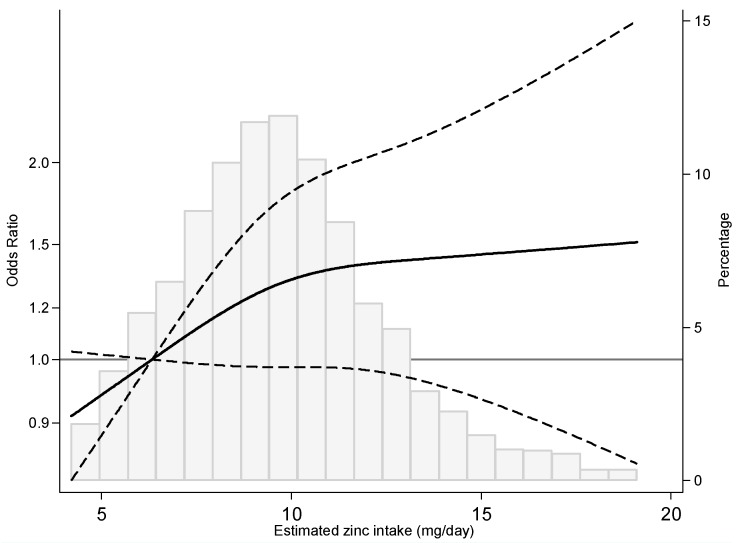
Prostate cancer odds ratio by estimated dietary zinc intake. OR (solid line) and 95% CIs (dashed lines) by estimated zinc intake modelled by using restricted cubic splines for zinc concentrations with knots at the 10th, 50th, and 90th percentile. The reference value is set at the 30th percentile of the zinc distribution. OR were adjusted for age, education, BMI, family history of prostate cancer, calcium intake and grains and legumes consumption as fixed effects, and province of residence as a random effect.

**Table 1 nutrients-11-00018-t001:** Characteristics of participants in case-control MCC-Spain study.

	All			Participants with Genetic Data		
	Controls	Cases	*p*	Controls	Cases	*p*
	*n* = 1228	*n* = 733		*n* = 821	*n* = 514	
**Zinc (mg/day) mean (sd)**	9.68 (2.94)	9.91 (2.98)	0.096	9.62 (2.91)	9.86 (2.94)	0.140
**Calcium (mg/day) median (IQR)**	876 (685;1117)	882 (671;1118)	0.858	864 (688;1126)	864 (657;1102)	0.596
**Energy intake (kcal/day) mean (sd)**	2013 (607)	2062 (611)	0.086	2000 (604)	2055 (615)	0.108
**Alcohol (g/day) median (IQR)**	18.9 (5.8;41.8)	22.0 (8.5;45.4)	0.010	20.3 (6.9;42.8)	22.9 (8.8;45.5)	0.073
**Grains & legumes (g/day) mean (sd)**	205 (84)	206 (80)	0.847	199 (81)	203 (77)	0.400
**BMI (kg/m2) mean (sd)**	27.48 (3.79)	27.65 (3.70)	0.339	27.55 (3.69)	27.53 (3.63)	0.945
**METs/week *n*(%)**			0.345^a^			0.241^a^
0 METs/week	495 (40%)	273 (37%)		304 (37%)	184 (36%)	
0.1–7.9 METs/week	158 (13%)	98 (13%)		118 (14%)	59 (11%)	
8.0–15.9METs/week	135 (11%)	96 (13%)		94 (11%)	73 (14%)	
≥16 METs/week	421 (34%)	266 (36%)		305 (37%)	198 (39%)	
Unknown	19 (2%)	0 (0%)		0 (0%)	0 (0%)	
**Smoking *n*(%)**			0.267^a^			0.552^a^
Never	324 (26%)	212 (29%)		218 (27%)	148 (29%)	
Former >1 year	636 (52%)	352 (48%)		416 (51%)	258 (50%)	
Smoker or former ≤1 year	266 (22%)	167 (23%)		186 (23%)	106 (21%)	
Unknown	2 (0%)	2 (0%)		1 (0%)	2 (0%)	
**Age (years) mean (sd)**	66.15 (8.58)	65.62 (7.31)	0.167	65.78 (8.21)	65.73 (7.35)	0.913
**Education *n*(%)**			0.000			0.001
No formal education	217 (18%)	163 (22%)		119 (14%)	103 (20%)	
Primary School	399 (32%)	285 (39%)		284 (35%)	204 (40%)	
Secondary School	344 (28%)	161 (22%)		231 (28%)	116 (23%)	
University or more	268 (22%)	124 (17%)		187 (23%)	91 (18%)	
**Familial history of PC *n*(%)**			0.000			0.000
No	1139 (93%)	580 (79%)		761 (93%)	410 (80%)	
2nd degree	15 (1%)	21 (3%)		8 (1%)	15 (3%)	
One of 1st degree	71 (6%)	113 (15%)		49 (6%)	76 (15%)	
More than one of 1st degree	3 (0%)	19 (3%)		3 (0%)	13 (3%)	
**Supplement intake *n*(%)**			0.214^a^			0.707^a^
None	1050 (86%)	695 (95%)		777 (95%)	487 (95%)	
Unspecified supplement	30 (2%)	12 (2%)		16 (2%)	8 (2%)	
Supplement with zinc	28 (2%)	24 (3%)		22 (3%)	17 (3%)	
Unknown	120 (10%)	2 (0%)		6 (1%)	2 (0%)	
**Gleason *n*(%)**						
Control	1228 (100%)			821 (100%)		
=6		333 (45%)			226 (44%)	
>6		388 (53%)			279 (54%)	
Unknown		12 (2%)			9 (2%)	
**Stage *n*(%)**						
Control	1228 (100%)			821 (100%)		
cT1-cT2a		578 (79%)			407 (79%)	
cT2b-T4		109 (15%)			83 (16%)	
Unknown		46 (6%)			24 (5%)	

^a^*p*-value excluding missing category. IQR: interquartile range; BMI: body mass index; MET: Metabolic Equivalent of Task; PC: prostate cancer.

**Table 2 nutrients-11-00018-t002:** Dietary zinc and prostate cancer risk in MCC-Spain case control study.

			Basic Model	Adjusted Model
	Controls	Cases	OR ^a^ (95%CI)	OR ^b^ (95%CI)
**ZINC**				
**T1 (<8.34 mg/day)**	409	217	1.00	1.00
**T2 (8.34–10.53 mg/day)**	409	245	1.12 (0.88;1.42)	1.17 (0.90;1.53)
**T3 (>10.53 mg/day)**	410	271	1.27 (1.00;1.61)	1.39 (1.00;1.95)
***p*-trend**			0.053	0.051

^a^ Odds Ratio (OR) of prostate cancer adjusted by age, education, BMI and family history of prostate cancer as fixed effects, and province of residence as a random effect. ^b^ Odds Ratio of prostate cancer adjusted by age, education, BMI, family history of prostate cancer, calcium intake and grains and legumes consumption as fixed effects, and province of residence as a random effect.

**Table 3 nutrients-11-00018-t003:** Association between dietary zinc and prostate cancer by tumor aggressiveness and extension.

		Gleason = 6 *n* = 333		Gleason >6 *n* = 388			cT1-cT2a *n* = 578		cT2b-T4 *n* = 109		

	**Co**	**Ca**	**RRR (95%CI)**	**Ca**	**RRR (95%CI)**	***p*-het**	**Ca**	**RRR (95%CI)**	**Ca**	**RRR (95%CI)**	***p*-het**
**ZINC**						0.177					0.754
**T1 (<8.34 mg/day)**	409	84	1.00	129	1.00		171	1.00	36	1.00	
**T2 (8.34–10.53 mg/day)**	409	116	1.39 (0.98;1.99)	124	1.03 (0.74;1.42)		195	1.18 (0.89;1.58)	33	0.95 (0.55;1.66)	
**T3 (>10.53 mg/day)**	410	133	1.66 (1.07;2.57)	135	1.22 (0.81;1.84)		212	1.40 (0.97;1.99)	40	1.24 (0.63;2.42)	
***p*-trend**			0.024		0.330			0.076		0.527	

Co: controls; Ca: cases; OR: Odds Ratio; CI: Confidence Interval; *p*-het: heterogeneity *p* value; RRR: relative risk ratio of prostate cancer adjusted by age, education, BMI, family history of prostate cancer, calcium intake and grains and legumes consumption as fixed effects, and province of residence as a random effect.

**Table 4 nutrients-11-00018-t004:** Association between dietary zinc and prostate cancer by genetic susceptibility to prostate cancer, measured with tertiles of the polygenic risk score (PRS).

	ALL (*n* = 514)			PRS2 T1 (*n* = 69)			PRS2 T2 (*n* = 170)			PRS2 T3 (*n* = 275)			
	Co	Ca	OR ^a^ (95%CI)	Co	Ca	OR ^b^ (95%CI)	Co	Ca	OR ^b^ (95%CI)	Co	Ca	OR ^b^ (95%CI)	*p*-het
**ZINC**													0.000
**T1 (<8.34 mg/day)**	272	158	1.00	94	30	1.00	85	45	1.00	93	83	1.00	
**T2 (8.34–10.53 mg/day)**	285	170	1.06 (0.77;1.46)	97	19	0.68 (0.34;1.35)	94	62	1.15 (0.68;1.93)	94	89	1.09 (0.69;1.73)	
**T3 (>10.53 mg/day)**	264	186	1.31 (0.88;1.95)	81	20	0.89 (0.43;1.85)	99	63	1.10 (0.62;1.97)	84	103	1.50 (0.89;2.53)	
***p*-trend**			0.182			0.698			0.751			0.119	

^a^ Odds ratio of prostate cancer adjusted by age, education, BMI, family history of prostate cancer, calcium intake and grains and legumes consumption as fixed effects, and province of residence as a random effect. ^b^ Odds ratio of prostate cancer adjusted by age, education, BMI, family history of prostate cancer, calcium intake and grains and legumes consumption as fixed effects and province of residence as a random effect, including an interaction with PRS (tertiles).

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
