# Peer review of "Dietary Zinc and Risk of Prostate Cancer in Spain: MCC-Spain Study"

_nutrients, 2018, doi:10.3390/nu11010018_

Round 1
Reviewer 1 Report
The authors present an analysis of dietary zinc intake and risk of prostate cancer using data from the MCC-Spain multi-center case control study. There are two major issues with the discussion/interpretation and conclusions of the results presented in this manuscript that dampen enthusiasm for this manuscript.
The lack of data on PSA screening is a major metholodologic issue in this study, and likely results in screening-related bias. As PSA screening is the strongest risk factor for prostate cancer, studies in populations not uniformly screened with PSA, as is the case with case-control studies of prostate cancer, (ie the majority of cases will have had a PSA, controls may not), are subject to detection bias.
Although the authors suggest in the last line of the discussion “a limiting point is our lack of good data on PSA screening, although it is very difficult to evaluate to what extent this factor relates to zinc intake in our participants”. It is well-understood that many ‘healthy lifestyle behaviors’ are related to the likelihood of getting a PSA test (1, 2). Thus, the distribution of these behaviors is likely to differ between cases and controls, and without accounting for differential PSA screening, the true association between a given lifestyle behavior (ie healthy diet/supplement use), is likely to be distorted. Indeed, the results presented, which show an increased odds of only ‘low-grade’/localized tumors, likely confirms the presence of this issue, as PSA screening leads to an increased detection of latent, lower-grade/stage cancers. If the authors have any data on PSA screening in this case-control study, I would strong urge them to consider it in their analyses.
The other issue is that this manuscript dismisses the inherent potential for differential bias in case-control studies of diet and cancer risk. Differential recall bias is a well-known and acknowledged epidemiologic issue that affects the validity of case-control studies of diet and prostate cancer. In Lines 349-350, the authors incorrectly suggest that recall bias is “hopefully” non-differential.
Other minor issues to consider:
Did the authors consider evaluating ‘total’ zinc as an exposure? Although zinc supplement use was not common, this would be comparable to the data presented by other studies.
Page 8, Lines 237-247: As you note, the HPFS cohort found an increased of zinc and prostate cancer, but need to clarify that this evaluation only looked at supplemental zinc (not zinc from dietary sources), and the association was only found for advance PCa – there was no association for organ-confined disease.
Did the authors consider a more stringent definition of high grade, as there is not strong consensus as to whether Gleason 7, at least primary pattern 3, is considered a ‘high grade’.
Lines 277-287: It’s very important that the authors note the low dietary zinc exposure in their study sample – however, what does it tell them about their “case-control” sample and the validity/generalizability of these results?
References:
1) Close DR, Kristal AR, Li S, Patterson RE, White E. Associations of demographic and health-related characteristics with prostate cancer screening in Washington State. CEBP 1996 Jul7(7):627-30.
2) Gonzalez A, Peters U, Lampe JW, White E. Zinc intake from supplements and diet and prostate cancer. Nutrition and Cancer 2009 Feb 61:2, 206-215,
Reviewer 2 Report
In this manuscript, a retrospective clinical study done to evaluate the correlation of dietary zinc and prostate cancer risk factors using a well-stablished and multi-central dataset named MCC-Spain. My first impression after reading the abstract of this manuscript was this study might have a low level of novelty since we have had same published investigations such as: Epstein, Mara M., et al. "Dietary zinc and prostate cancer survival in a Swedish cohort" The American journal of clinical nutrition 93.3 (2011): 586-593. However, after reviewing the results and the provided comprehensive discussion, I believe this paper fulfils the requirements to be published as an original article. Particularly, this manuscript demonstrated that the differential effect of zinc intake proportional with the genetic susceptibility cases.
Author Response
We want to thank the reviewer for his/her appreciative comments and interest.
Round 2
Reviewer 1 Report
The authors have addressed my concerns.